



# GEOS-Chem High Performance (GCHP): A next-generation implementation of the GEOS-Chem chemical transport model for massively parallel applications

Sebastian D. Eastham[1,2]*, Michael S. Long[2], Christoph A. Keller[3,4], Elizabeth Lundgren[2], Robert M. Yantosca[2], Jiawei Zhuang[2], Chi Li[5], Colin J. Lee[5], Matthew Yannetti[2], Benjamin M. Auer[3,6], Thomas L. Clune[3], Jules Kouatchou[3], William M. Putman[3], Matthew A. Thompson[3], Atanas L. Trayanov[3], Andrea M. Molod[3], Randall V. Martin[5,7], Daniel J. Jacob[2]

[1]Laboratory for Aviation and the Environment, Massachusetts Institute of Technology
[2]John A. Paulson School of Engineering and Applied Sciences, Harvard University
[3]NASA Global Modeling and Assimilation Office
[4]Universities Space Research Association, Columbia, MD, USA
[5]Department of Physics and Atmospheric Science, Dalhousie University
[6]Science Systems and Applications, Inc., Lanham, MD, USA
[7]Smithsonian Astrophysical Observatory, Harvard-Smithsonian Center for Astrophysics

*Correspondence to*: Sebastian D. Eastham (seastham@mit.edu)

**Abstract.** Global modeling of atmospheric composition is a grand computational challenge because of the need to simulate large coupled systems of chemical species interacting with transport on all scales. Off-line chemical transport models (CTMs), where the chemical continuity equations are solved using meteorological data as input, have the advantages of simplicity and reproducibility, and are important vehicles for developing knowledge that can then be transferred to Earth system models. However, they have generally not been designed to take advantage of massively parallel computing architectures. Here we develop such a high-performance capability (GCHP) for GEOS-Chem, a CTM driven by GEOS meteorological data from the NASA Goddard Earth Observation System (GEOS) and used by hundreds of research groups worldwide. GCHP is a grid-independent implementation of GEOS-Chem using the Earth System Modeling Framework (ESMF) that permits the same standard model to be run in a distributed-memory framework, scalable from six cores on a single node up to hundreds of cores distributed across a network. GCHP also allows GEOS-Chem to take advantage of the native GEOS cubed-sphere grid for greater accuracy and computational efficiency in simulating transport. GCHP enables GEOS-Chem simulations to be conducted with high computational scalability up to at least 500 cores, so that global simulations of stratosphere-troposphere oxidant-aerosol chemistry at C180 spatial resolution (~0.5°×0.625°) or finer become routinely feasible.



# 1 Introduction

Atmospheric chemistry models are used to address a wide range of problems related to climate forcing, air quality, and atmospheric deposition. Simulations of oxidant and aerosol chemistry involve hundreds of chemically interacting species, coupled to transport on all scales. The computational demands are considerable, which has limited the inclusion of

atmospheric chemistry in climate models (National Research Council, 2012). Off-line chemical transport models (CTMs), where meteorology is provided as input data from a parent global climate model (GCM) or atmospheric data assimilation system (DAS), are frequently used for reasons of simplicity, reproducibility, and ability to focus on chemical processes. The global GEOS-Chem CTM originally described by Bey et al. (2001), using meteorological input from the Goddard Earth Observation System (GEOS) DAS of the NASA Global Modeling and Assimilation Office (GMAO), is used by hundreds of

atmospheric chemistry research groups worldwide (http://www.geos-chem.org). Increasing computational resources in the form of massively parallel architectures can allow GEOS-Chem users to explore more complex problems at higher grid resolutions, but this requires re-engineering of the model to take advantage of these architectures. Here we describe a high-performance version of GEOS-Chem (GCHP) engineered for this purpose, and we demonstrate its ability to access a new range of capability and scales for global atmospheric chemistry modeling.

The original GEOS-Chem CTM ("GEOS-Chem Classic", or GCC) was designed for shared-memory (OpenMP) parallelization. Computation is distributed over a number of cores on a single node, with data held in shared arrays. But recent growth in computational power has taken the form of massively-parallel networked systems, where additional computational power is achieved by increasing the number of identical nodes rather than by improving the nodes themselves.

This has placed a restriction on growth in the problem size and complexity which can be solved by a single instance of GCC. To take advantage of massively-parallel architectures, a new framework is needed which allows GEOS-Chem to use a distributed-memory model, where the computation is distributed across multiple coordinated nodes using a Message Passing Interface (MPI) implementation such as MVAPICH2 or OpenMPI.

An important first step in this evolution was the integration of GEOS-Chem as the online chemistry component within the GEOS DAS (Long et al., 2015). In order to ensure that the online and offline versions of GEOS-Chem were identical, GCC was modified so that the exact same code is now used in the independent CTM and in the DAS. Major modifications were required to make GEOS-Chem grid-independent and compatible with the GEOS Modeling and Analysis Prediction Layer (MAPL) (Suarez et al., 2007), an Earth System Modeling Framework (ESMF) (Hill et al., 2004) based software layer which

handles communication between different components of the GEOS DAS. The GEOS-Chem code was adapted to accept an arbitrarily-sized horizontal set of atmospheric columns, with no requirements regarding adjacency of the columns or overall coverage of any particular set. All these changes were made "under the hood" in the standard GEOS-Chem Code. When GEOS-Chem is run as GCC, the set of columns is designated as a single block which covers the entire globe or a subset in a



nested domain, and parallelization is achieved by internally running parallel loops over the columns. When GEOS-Chem is run as part of GEOS, MAPL internally splits the atmosphere into smaller domains, each of which contains a different set of atmospheric columns. These domains can then be distributed across multiple nodes, exploiting massively-parallel architectures. As a result of these changes, the same GEOS-Chem code can now be run either as a stand-alone, shared-memory offline CTM, or as a GCM component in the massively-parallel, distributed GEOS DAS. Any improvement in chemical modeling developed for the offline CTM is thus immediately available in the GEOS DAS version, which never becomes out of date and remains referenceable to the current version of the model.

In this work we take the next logical step of developing GCHP as a distributed-memory, MAPL-based implementation of the GEOS-Chem CTM. GCHP uses an identical copy of the GEOS-Chem Classic (GCC) code to provide the same high-fidelity atmospheric chemical simulation capabilities, allowing users to switch between GCC and GCHP implementations with confidence that they are using the same model. The exact same internal code is used in GCC shared-memory and GCHP distributed-memory applications. This closes the development loop between online and offline modeling. By sharing infrastructure code between GCHP and GEOS in the form of MAPL, offline modelers can now take advantage of modeling advances which originate in the online model in the same way that GEOS benefits from advances in chemical modeling developed in the GEOS-Chem CTM (Nielsen et al., 2017). By way of example, GEOS was recently able to conduct a full-year 13-km resolution "nature run" with the current standard version of GEOS-Chem tropospheric chemistry (Keller et al., 2017). In return, GCHP incorporates the more efficient cubed sphere grid and FV3 advection code present in GEOS, and is capable of directly ingesting GEOS output in its native cubed sphere format.

## 2 Model description

### 2.1 Overview

Atmospheric chemistry models such as GEOS-Chem solve the 3-D chemical continuity equations for an ensemble of $m$ coupled chemical species (Brasseur and Jacob, 2017). The continuity equation for the number density $n_i$ [molecules cm$^{-3}$] of species $i$ is expressed as

$$\frac{\partial n_i}{\partial t} = -\nabla \cdot (n_i \mathbf{v}) + s_i \tag{1}$$

where $\mathbf{v}$ is the velocity vector [m s$^{-1}$], and $s_i$ is the local net production and loss of species $i$ [molecules cm$^{-3}$ s$^{-1}$]. In CTMs, $\mathbf{v}$ is provided by archived output from a parent GCM or DAS, with subgrid-scale parameterized transport statistics (boundary layer mixing, deep convection) as additional CTM transport terms in Eqn. 1. From a computational standpoint, the local term $s_i$ is grid-independent. However, the transport terms are grid-aware, as they move material between grid points. In GEOS-Chem, the atmosphere is split into independent columns, with each column made up of a number of discrete grid points (Long et al., 2015). Vertical processes (boundary layer mixing, deep convection) are then considered to be local in the sense



that they are calculated independently for each column. In each column simulated by GEOS-Chem, the local term computes chemical evolution with a unified tropospheric-stratospheric mechanism (Eastham et al., 2014; Sherwen et al., 2016), convective transport (Wu et al., 2007), boundary layer mixing (Lin and McElroy, 2010), radiative transfer and photolysis (Prather, 2012), wet scavenging (Liu et al., 2001), dry deposition (Wang et al., 1998), particle sedimentation (Fairlie et al., 2007), and emissions (Keller et al., 2014).

The GEOS DAS meteorological fields used for input to GEOS-Chem are produced on a gnomonic cubed sphere grid (Putman and Lin, 2007) at a current horizontal resolution of C720 (~13 km × 13 km), with output provided operationally on a rectilinear grid at a resolution of 0.25°×0.3125°. Currently, global oxidant-aerosol simulations with GEOS-Chem are effectively limited to 2°×2.5° resolution due to the prohibitive memory and time requirements of running a more finely-resolved simulation on a single node. In order to progress to finer resolutions, GEOS-Chem must be able to split the requirements for memory and computation across multiple nodes, and to ensure that communication between the different nodes is minimal and efficient. This is the role of GCHP.

## 2.2 GCHP model architecture

The general software architecture of the GCHP model is shown in Figure 1. The GMAO-developed MAPL is included in the GCHP code download and is automatically built when compiling GCHP for the first time. MAPL initializes the model, establishes the atmospheric domain on each computational core, and handles model coordination and internal communication. Transport within and between each of the domains is calculated by the FV3 advection component. Within each atmospheric domain, local terms are calculated by a standard copy of the GEOS-Chem Classic code, embedded in the model as described by Long et al. (2015). Data input is handled through the External Data (ExtData) component, and output is handled through the History component. ExtData and History are structural components of MAPL (Long et al., 2015; Nielsen et al., 2017; Suarez et al., 2007).





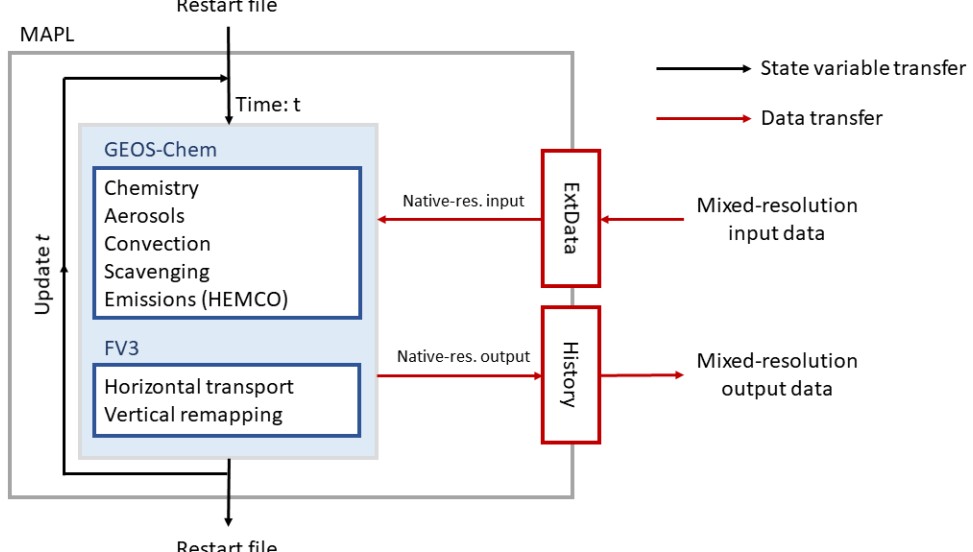

**Figure 1. Connectivity of the major components of GCHP. The main time stepping loop is represented by the feedback loop from the model output back into the input.**

At initialization, a gridded representation of the atmosphere is generated by MAPL from user-specified input. GCHP can

operate on any horizontal grid supported by MAPL as long as an appropriate advection scheme is available. Currently, the standard advection scheme in GCHP is the Putman and Lin FV3 scheme, which operates on a cubed sphere discretization, described in section 2.2. The initial state of the model is determined from a restart file, read by MAPL directly. During this stage, all relevant input data are also read into memory through the ExtData module. Data at any grid resolution are read from NetCDF files in disk storage, and are regridded on the fly to the resolution at which the model is running. This allows

data on either rectilinear latitude-longitude or gnomonic cubed sphere grids to be read in without requiring offline preprocessing. Data can be regridded using bilinear interpolation (used for wind fields), or first-order mass-conservative regridding (used for emissions and all other meteorological data). Conservative regridding is achieved using "tile files" generated analytically with the Tempest tool (Ullrich and Taylor, 2015). Wind data are regridded by bilinear interpolation of the wind vector. Additional regridding techniques are also available for special cases such as handling categorical (e.g.

surface type) data. All constant fields are read in once, at the start of the simulation. For all time-varying fields, ExtData holds two samples in memory at all times: the previous sample ("left bracket") and the upcoming sample ("right bracket"). All fields can either be held constant between samples, or smoothly interpolated between the two brackets.

Output is performed through the History component. Fields which are defined as "exports" within GCHP are tracked

continuously by the History component. Any export can be requested by the user by adding it to an output collection in the HISTORY.rc input file, as either an instantaneous and/or time-averaged output. At each time step, the History component will acquire the current value of the field for each requested diagnostic and store it either at the native resolution or, if



requested by the user, perform online regridding to a rectilinear latitude-longitude grid. This allows the user to decide the appropriate spatial and temporal resolution for their simulation output, independent of the resolution at which the simulation itself is conducted. All diagnostic quantities which are available in gridded form in GEOS-Chem Classic are automatically defined as exports in GCHP.

## 2.3 Grid discretization and transport

In GCHP the atmosphere is divided into independent atmospheric columns, with a subset of columns forming a single domain which is assigned to one of the computational cores. All local operations, such as chemistry, deposition, and emissions, are handled locally by components already present in the core GEOS-Chem code. The advection operator transfers mass between adjacent columns, requiring MPI-based data communication between them at domain boundaries. The amount and frequency of the communication depends on the chosen grid discretization and transport algorithm.

### 2.3.1 Grid discretization

GCHP inherits the equidistant gnomonic cubed sphere grid discretization used by the GEOS DAS (Putman and Lin, 2007). Cubed sphere grids split the surface of a sphere into six equal-sized faces. Each face is then subdivided into cells of approximately equal size, with each cell representing an atmospheric column. The equidistant gnomonic projection splits each cube edge into $N$ equally-sized segments, connecting the opposing edges with great circle arcs in order to generate a regular mesh (see Figure 2). The grid resolution is referred to as C$N$, such as C48 for a grid with 48×48 atmospheric columns on each of the 6 faces. The grid cell spacing is approximately 10,000/$N$ km, such that a C48 grid has a mean cell width of ~200 km. Each core is assigned a contiguous, rectangular set of columns on one of the 6 faces by MAPL, with the exact subdomain size determined based on the domain aspect ratio specified by the user at run time.





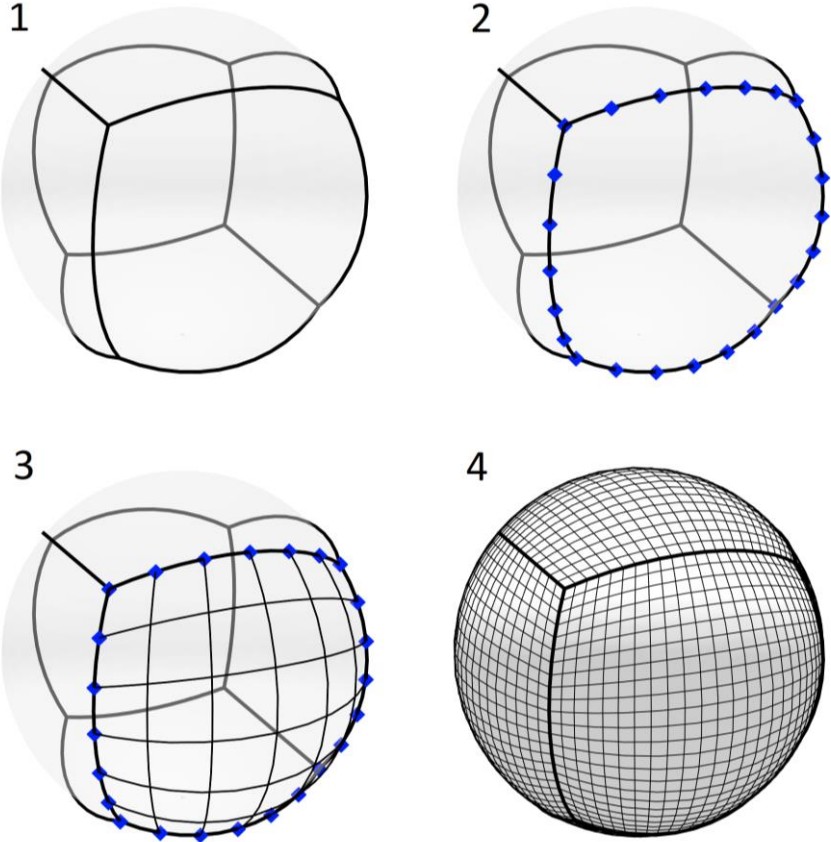

**Figure 2. Graphical description of the process used to generate a gnomonic cubed sphere grid. Sub-panels are numbered based on the textual description of the steps. The grids shown are C6 and, on the final frame, C24. Demonstration is available interactively at http://www.geos-chem.org/cubed_sphere.html.**

5    Cubed sphere grids offer several advantages over conventional rectilinear grids. The absolute cell size in a rectilinear grid decreases from the equator to the poles, for structural rather than scientific reasons, resulting in larger Courant-Friedrichs-Lewy (CFL) numbers at high latitudes. This reduces the minimum time step required for explicit Eulerian advection schemes to maintain stability. The problem can be mitigated by applying a semi-Lagrangian method when the CFL exceeds unity, at the expense of additional computational overhead, inconsistency in technique, and difficulties in ensuring mass conservation.

10   In an MPI environment, these issues also complicate domain decomposition for the purposes of distributing the grid between cores. The use of a semi-Lagrangian scheme results in tracer mass being transferred between grid cells which are not considered to be adjacent, increasing the amount of communication necessary between cores. Furthermore, any subdomain with an edge at a pole will be required to communicate with all other cores sharing the pole.



The cubed sphere grid helps to address these issues. The area ratio between the largest and smallest grid cells is ~2.3, regardless of the resolution. There are no polar singularities, although advection across the edges and corners of the cube requires special considerations.

Vertically, the atmosphere is discretized into a series of hydrostatic, hybrid-sigma layers. The current GEOS DAS uses 72 layers ranging from the surface to 1 Pa at the upper edge.

### 2.3.2 Transport

Transport in GEOS-Chem is comprised of four operations: advection, moist convection, boundary-layer mixing, and aerosol settling. The latter three operations occur purely in-column and are unchanged between GCC and GCHP. However,

advection must be grid-aware. Horizontal advection in GCHP is calculated on a layer-by-layer basis using the cubed sphere advection algorithm of Putman and Lin (2007). This algorithm is 4th-order accurate except at the six cube edges (2nd-order). Vertical advection is then calculated using a vertically-Lagrangian method (Lin, 2004). Prior to the advection step, each core requests information from neighboring domains to provide information on boundary conditions. Advection is then calculated independently for each atmospheric domain.

Horizontal mass fluxes and CFL numbers are either supplied directly to the model or are calculated based on 3-hour average horizontal wind speed data and the instantaneous surface pressure at the start of the time step (time $t$). All fluxes are based on dry air mass and dry surface pressure/ To ensure numerical stability, sub-stepping is implemented such that the internal advection timestep $\Delta t$ is sub-divided into $n$ number of sub-steps until the CFL is less than one. Horizontal advection is then

performed $n$ times. If the number of sub-steps $n$ is greater than 1, changes to the air mass in each grid cell due to wind divergence are retained between sub-steps. The implied surface pressure resulting from changes in total column mass is also updated. However, horizontal mass fluxes are assumed to be constant over the time step, and no vertical remapping is performed between sub-steps. When mass fluxes have to be estimated offline from wind data, the simulated pressure can diverge from that in the meteorological archive (Jöckel et al., 2001). GCC solves this problem with the pressure fixer of

Horowitz et al. (2003), which modifies calculated air mass fluxes to ensure the correct surface pressure tendency based on zonal totals. However, this approach corrupts the horizontal transport to some extent. GCHP defaults instead to a simple global air mass correction also applied to tracers.

After the horizontal tracer advection loop is complete (time $t+\Delta t$), the total air mass in each vertical column will have

changed, as will the vertical distribution. Vertical advection is calculated by remapping the deformed layers back to the hydrostatic hybrid-eta grid defined by the surface pressure, as interpolated from the meteorological archive for the post-advection time ($t+\Delta t$). This ensures that the surface pressure accurately tracks that in the meteorological data.





## 2.4 Benchmarking

The standard benchmarking procedures applied to GEOS-Chem before each version release are also applied to GCHP, ensuring that the integrity of the model is maintained from version to version. Benchmarks involve a 1-year UCX (troposphere-stratosphere) oxidant-aerosol simulation with resolution of 4ºx5º (GCC) or C48 (GHP), plus a 1-year

simulation of the $^{222}$Rn-$^{210}$Pb-$^{7}$Be system (Liu et al., 2001) for updates that may affect transport. Further documentation of benchmark procedures is available at http://www.geos-chem.org. Species concentrations and source/sink diagnostics from the benchmark simulation are archived and compared to the previous model version and to selected climatological data. Results are inspected by the model developers and by the GEOS-Chem Steering Committee, which gives final approval. There are small differences between GCHP and GCC benchmarks that we can relate to differences in transport algorithm,

but otherwise the two functionalities perform identically.

## 3 Model performance

We analyzed the performance of GCHP by conducting simulations multiple grid resolutions (C24 to C180), each for a range of core counts (Table 1). For low-resolution applications, performance is also compared to the maximum achievable performance using the GCC shared-memory architecture. All simulations are for 1 month (July 2016) of troposphere-

stratosphere oxidant-aerosol chemistry, including 206 species and 135 tracers, and using operational meteorological data from GEOS FP. The GCC simulations use previously-regridded 2°×2.5° and 4°×5° meteorological fields, while the GCHP simulations regrid the 0.25°×0.3125° fields to the cubed sphere on the fly through ExtData. A native-resolution cubed sphere GEOS data output stream is presently under development at GMAO and will benefit GCHP by reducing the need for regridding.




**Table 1. Grid resolution and core counts for the performance test simulations.**

| GEOS-Chem implementation | Grid resolution | Number of grid cells | Number of cores used |
|---|---|---|---|
| GCHP | C24 | 250,000 | 6 – 216 |
| GCHP | C48 | 1,000,000 | 6 – 540 |
| GCHP | C90 | 3,500,000 | 12 – 540 |
| GCHP | C180 | 14,000,000 | 90 – 540 |
| GCC | 4°×5° | 240,000 | 6 – 30 |
| GCC | 2°×2.5° | 940,000 | 6 – 30 |

All simulations were conducted on the Harvard Odyssey computational cluster. Both GCHP and GCC were compiled using the Intel Fortran compiler (v15.0.0), and MPI capabilities for GCHP were provided by OpenMPI (v1.10.3). Each node of the

5    cluster has 32 Intel Broadwell 2.1 GHz cores sharing 128 GB of RAM, and all nodes are connected via Mellanox FDR Infiniband. Input and output data are stored using a Lustre parallel file system, accessible through the same Infiniband network fabric. All simulations were scheduled to enforce exclusive access to the nodes, preventing possible performance degradation due to sharing of node resources.

10   Figure 3 shows the total time taken to perform the simulation at each resolution, both in terms of wall time and in terms of the time per 1,000 simulated atmospheric columns in the model grid. Results using the conventional GEOS-Chem Classic (GCC) shared-memory platform at 4°×5° and 2°×2.5° are also shown for comparison.





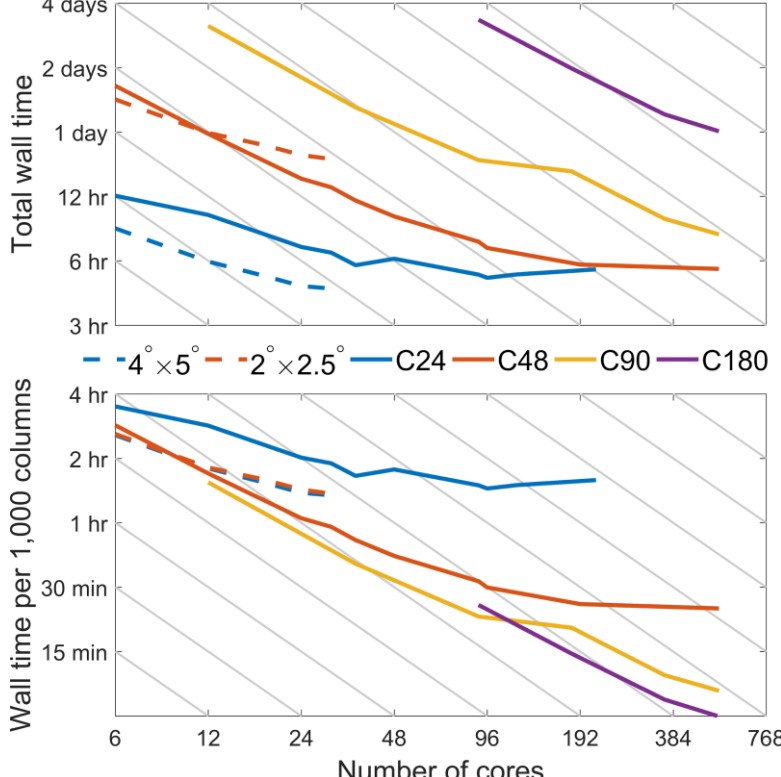

**Figure 3. Wall time taken to perform 1-month GEOS-Chem simulations at different resolutions and with different numbers of cores. The upper panel shows the absolute time taken to complete each simulation at each resolution. The lower plot shows the wall time normalized by the number of atmospheric columns simulated at each resolution. Solid lines are for GCHP simulations (cubed sphere grids) and dashed lines are for GCC simulations (latitude-longitude grids). Grey lines on each plot show perfect scaling, corresponding to a 50% reduction in run time for each doubling of the number of cores.**

At the lowest simulated resolution (C24), GCHP's runtime exceeds that of GEOS-Chem Classic. This is predominantly due to overhead associated with file open operations, as the native-resolution meteorological data used to drive GCHP are opened and read independently for each field. This can be addressed in the future through both structural changes and parallelization of the input.

After doubling the resolution to C48 (2°×2.5°), GCHP begins to out-perform GCC, with a reduced overall simulation time even at core counts which are currently accessible to GCC, despite the larger input requirements of GCHP. A 1-month simulation at C48 requires only 6 hours using 96 cores. GCHP scalability also improves as the model resolution increases. At C180, the reduction in simulation time for each doubling in the number of cores is approximately a factor of 1.6.

Two factors affect GCHP scalability: fixed costs, and overhead. Fixed costs are for operations which run on a fixed number of cores, regardless of the number of cores being dedicated to the simulation. Overhead is the need for additional coordination and data communication between cores, which grows with the number of cores. Eventually the overhead of the




additional cores exceeds the computational benefit, resulting in a performance plateau. This overhead includes one-off costs, such as the MPI interface initialization, which can be significant when running with a large number of cores but which can be reduced in relative terms by running longer simulations.

5    The scalability of each model component is shown in Figure 4. This shows the total time spent on each component at C48 and C180 resolution as a function of the total number of cores used, from 6 cores up to 540. We see that the dominant fixed cost at both C48 and C180 is input, which also dominates the overall cost for C48 with more than 48 cores. This is due to the serial nature of the current input code, overhead associated with file open operations, and the aforementioned use of native-resolution meteorological data for even low-resolution GCHP simulations. Output operations are a second fixed cost, being 10   handled by a single core at all times. For these simulations, 22 3D fields were stored with hourly frequency, and output was a minor contributor to overall costs. Fixed costs can be converted into scalable costs by parallelizing the component in question, and this is a future work agenda.

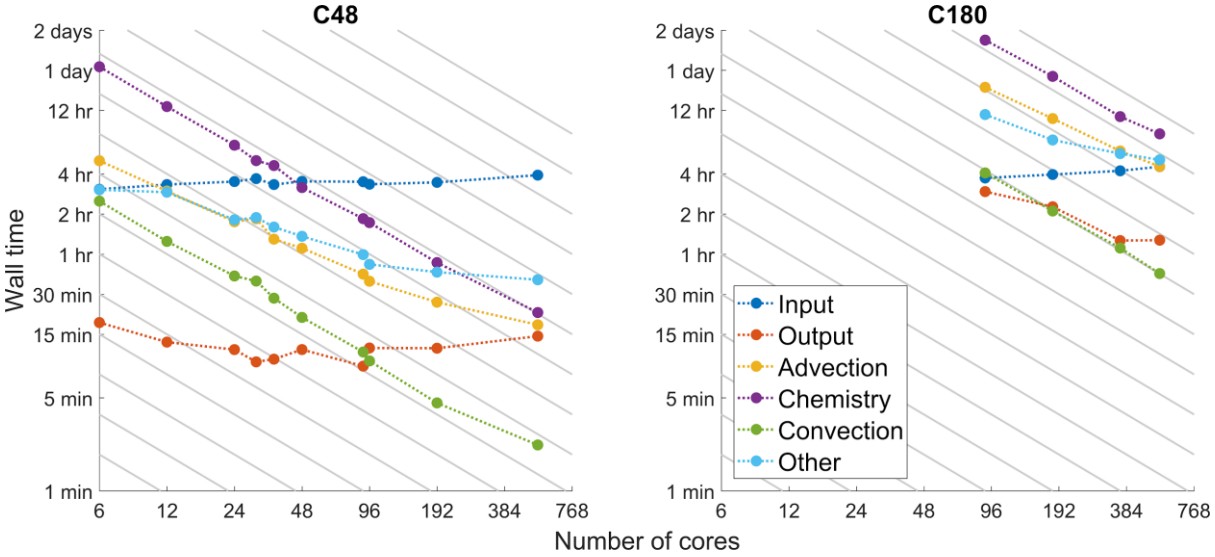

**Figure 4. Total wall time per component for a low (C48) and high (C180) resolution simulation. Simulations at C180 are limited to core counts of 90 or more across several nodes for the hardware used here, due to the high memory requirements of such high-resolution simulations.**

Chemistry, advection, and convection all scale well with increasing core count. Chemistry is the most expensive process at both coarse (C48) and fine (C180) resolution, but has near-perfect scalability. Thus at C48 we see that input becomes the 20  limiting process when the number of cores exceeds 48. Advection and other processes show more departure from perfect scalability, and may dominate the time requirement as the number of cores exceeds 600. The scalability of advection suffers from the additional communication overhead associated with reducing the domain size, as each domain must communicate a larger proportion of its concentration data to its neighbors. However, wall time for advection does consistently fall with



increasing core counts, an improvement compared to Long et al. (2015) where wall time increased as core counts exceeded 200 for a grid resolution equivalent to C48. We attribute this to the change from a latitude-longitude grid to the more scalable cubed sphere grid.

The remaining wall time is taken up by the "other" component, a mix of scalable and non-scalable processes. This includes the one-off cost of initializing the MPI interface, which grows non-linearly with the number of cores. At C180, these costs are still exceeded by scalable costs when running with 540 cores, so no plateau in performance is observed.

## 4 High resolution simulations with GCHP

The primary advantage of GCHP is the ability to perform high-resolution atmospheric modeling at resolutions previously not
available to the community. Figure 5 shows illustrative distributions of simulated ozone concentration at 4 km altitude and aerosol optical depth (AOD), simulated at C24 and C180. Simulations were performed using 24 and 360 cores respectively.

Global-scale patterns in ozone concentration and AOD are not substantially affected by the increase in resolution, demonstrating the ability of GCHP to accurately simulate synoptic scale chemistry and transport at even low resolutions.
However, increasing the horizontal resolution improves the ability of the model to capture the behavior of intercontinental plumes (Eastham and Jacob, 2017). The consequences of this are visible in the ozone distributions over the Pacific and Atlantic. We also observe maxima in the coarse-resolution simulation which are not visible in the finer-resolution simulation, such as the peak in ozone concentration over Egypt. This suggests possible simulation biases at coarse resolution. Increasing the horizontal resolution also improves the ability of the model to resolve features at the scale of local air quality.
This is especially evident in the simulated column AOD over northern India and Beijing.





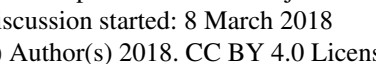

**Figure 5. Model output from GCHP simulations at C24 (left) and C180 (right) for July 31$^{st}$, 2016 after 1 month of initialization. The upper panels show 23:00 GMT ozone concentrations at 4 km altitude, and the lower panels show daily-average column aerosol optical depth (AOD) with each major aerosol class in GEOS-Chem represented using a different color. Calculated values in some regions exceed the displayed limits. Zoom panels are also shown for ozone over Europe and AOD over East Asia.**



## 5 Summary

Models of atmospheric chemistry have grown continuously in resolution and complexity over the past decades to take advantage of increasing computational resources. The GEOS-Chem High Performance model (GCHP) is a next step in this growth, enabling the widely-used GEOS-Chem chemical transport model to exploit the computational speed and memory

capacity of massively parallel architectures. In this manner we can achieve routine simulation of global stratosphere-troposphere oxidant-aerosol chemistry at unprecedented resolution and detail.

GCHP incorporates an unchanged copy of the existing GEOS-Chem shared-memory code into an ESMF-based framework (MAPL), enabling GEOS-Chem to be run in a distributed memory framework across multiple nodes while retaining all the

features of the high-fidelity global chemical simulation. In addition to a new model framework, GCHP replaces the conventional rectilinear latitude-longitude grid with a gnomonic cubed sphere grid. This provides greater computational accuracy and efficiency for transport calculations while removing an additional restriction on scalability. GCHP is scalable from six cores up to at least 540, completing a 1-month simulation of oxidant-aerosol chemistry in the troposphere and stratosphere (206 active species, 135 tracers) at a global resolution of C180 (~0.5°×0.625°) in 24 hours.

GCHP also provides a mechanism for ongoing improvement of modeling capability. With the base GEOS-Chem model, GCHP, and the GMAO GEOS atmospheric data assimilation system now all using an identical copy of the grid-independent GEOS-Chem code, GCHP closes the loop between online and offline modelers, allowing seamless propagation of model and framework improvements between all three. Future development opportunities range from improved parallelism in input

operations to the direct ingestion of archived mass fluxes to further improve transport calculation accuracy.

## Code availability

GCHP has been openly available as part of the GEOS-Chem code since beta version release v11-02b in June 2017, and will be part of the v11-02 public release in March 2018. Complete documentation and access to the GCHP code can be found at http://www.geos-chem.org. GCHP is an added functionality for GEOS-Chem users, who can choose to use either GCC or

GCHP from the same code download.  Both GCC and GCHP functionalities will be maintained in the standard GEOS-Chem model for the foreseeable future, recognizing that many users may not have access to the resources needed to use GCHP.

## Acknowledgements

This work was supported by the NASA Atmospheric Composition Modeling and Analysis Program (ACMAP) and the NASA Modeling, Analysis and Prediction (MAP) program. Sebastian D. Eastham was supported by the NOAA Climate and

Global Change Postdoctoral Fellowship Program, administered by UCAR's Visiting Scientist Programs. Sebastian D.



Eastham was also supported by a Harvard University Center for the Environment (HUCE) Postdoctoral Fellowship. The GEOS FP data used in this study/project were provided by the Global Modeling and Assimilation Office (GMAO) at NASA Goddard Space Flight Center. The computations were run on the Odyssey cluster supported by the FAS Division of Science, Research Computing Group at Harvard University. We are grateful to Compute Canada for hosting the Data Portal for

GEOS-Chem to store and make available the GEOS meteorological fields used here. The authors would also like to thank Kevin Bowman for providing analytical insight and additional computational resources with which to perform early testing of the GCHP model. Finally, we would like to thank Junwei Xu for assistance in processing and archiving meteorological data for input to the model.

**Author contribution**

MSL and AM designed the initial code infrastructure

RVM and DJJ provided project oversight and top-level design

SDE, MSL, CAK, EWL, MY, RY, JZ, MT, AT, TC, JK, WP, and CJL performed code development

SDE, EL, RVM, CL, and CJL adapted the meteorological archive

SDE, MSL, EWL, and JZ ran and debugged benchmark simulations

SDE, MSL, RY, and JZ performed scalability testing analysis

SDE, MSL, CAK, RY, EWL, JZ, RVM, and DJJ wrote the manuscript

All authors contributed to manuscript editing and revisions

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
