# Peer review of "GEOS-Chem High Performance (GCHP v11-02c): A next-generation implementation of the GEOS-Chem chemical transport model for massively parallel applications"

_Geoscientific Model Development, 2018_

## Short Comment (SC1) · 20 Mar 2018

Dear authors,

1. Please include the version number for GCHP in the title, and throughout the manuscript.
2. The precise version of the code discussed in the manuscript must be available. The current best practice is for this code to be uploaded to a public repository and a DOI assigned. The DOI should be cited in the manuscript.

[Figure]

Astrid Kerkweg

---

## Referee Comment (RC1) · Anonymous Referee #1 · 23 Apr 2018

**Title:** GEOS-Chem High Performance (GCHP): A next-generation implementation of the GEOS-Chem chemical transport model for massively parallel applications

**Authors:** Sebastian D. Eastham et al.

**Journal:** GMD

**Ref.:** gmd-2018-55 (MS Type: Model description paper)

**Summary**

I am not sure what to expect from these sort of papers as they do describe complex systems with only couple of pages of words. So if you are an outside reader then it is almost impossible to get a firm idea about the described system. It is also debatable whether these brief descriptions have any usefulness to other researcher? The model described in this paper has mainly two components, (i) a chemistry module, which is local and should, in principle, have a near perfect scaling (without any MPI instruction), while the second component (ii) is advection, which is not local and requires halo filling (or communication on demand) for parallel implementations. The only possible reason for publishing such papers would be to provide a reference for the model in question.

**Detailed comments:**

1. I think the paper needs a table summarizing the definitions of all the acronyms (GEOS-Chem, CTM, GCHP, ect...) used, because I kept going forward and backward to look for their meanings in the text.

2. My understanding of atmospheric chemistry models is that they solves a system of coupled ODEs at each grid of the model and each grid point is, in principle, independent of its neighbors. In other words, the chemical model needs information on the grid only and therefore such process should scale perfectly and these models are ideal for parallelism. However, advection needs information about the characteristic (hyperbolic problems) and this part that needs effort to make it work with parallel implementation (MPI). This looks like a coding task of combining FV3 transport (Lin et al.) with an existing chemistry model (with OpenMP). I think, the whole description of the new system and how its differs from the original GCC could be improved. There should be more clarifications and a detailed description of what changed from the original code (probably a table listing all the components of GCHP and how it differs from the original GCC).

3. Page 1, Abstract, Line 2. I am not sure how large the system of chemical species is? It would be better to give an order of unknowns for a typical system, i.e., order 100-1000000?.

4. The discussion at page 7 in relation to semi-Lagrangian, cube-sphere and lat-long grids is a bit subjective!

    (a) I am not sure what the authors means by "inconsistency in technique" in the sentence "The problem can be mitigated ......in ensuring mass conservation"?

    (b) I think the whole passage "In an MPI environment, ..... sharing the pole." cannot be justified. Semi-Lagrangian (SL) schemes on any grid with the right halo size can achieve near perfect scaling [see for example with up to 100000 cores SL scalability experiments in the paper: *High-performance high-resolution semi-Lagrangian tracer transport on a sphere, J.B. White III and J.J. Dongarra, Journal of Computational Physics, Vol. 230, pp. 6778–679 (2011)*] or up to 10000 cores scalability results in *Allen and Zerroukat, Journal of Computational Physics, Vol. 319, p. 44-60 (2016)]*. Certainly the only argument against SL is the lack of inherent conservation, but in terms of accuracy, stability, simplicity, computational cost, and scalability there is no valid arguments against SL.

    (c) Of course there is no perfect scheme and every approach has some advantages and disadvantages and one has to be objective about these things. For example the cube-sphere has the disadvantages of the complexity of dealing with 6 panels and their orientations and the non-orthogonality of the grid and its associated grid-imprinting (for example see your figure 5 for C24 where it is very clear that the pattern of the tracer distribution is very much influenced by the grid). It would be interesting to show the equivalent pictures (blowup pictures at Figure 5 for C24) due to the original GCC!.

5. Page 8, 1st paragraph of section 2.3.2. In parallel domain decomposition, I would suggest to use the right nomenclature of halo-filling instead of "providing boundary conditions", because there is no real boundaries between processors.

6. Page 8, end of 2nd paragraph of section 2.3.2. It is not clear what does mean the comment: "GCHP defaults instead to a simple global air mass correction also applied to tracers"?? My understanding is that the main purpose of using FV3 is to achieve mass conservation in an inherent way so I don't understand what is this correction is doing? Does it mean the inherent conservation is lost? Further clarifications are needed here.

7. I don't see any use for to the bottom sub-figure (time/1000colums) of figure 3.

8. Section 4 (example simulation). It is good to show how the scalability of GCHP can be exploited to run higher-resolution simulations and the improvement that comes as results of that (Figure 5). However, I feel there are some figures missing in this section/paper. Although there is a lot figures comparing GCC and GCHP in terms of computational efficiency (scaling and wall clock time), for fidelity there should be at least a couple of figures comparing (at least visually) GCC and GCHP for a benchmark test to show that the solutions of the two models are, at least visually, in a good agreement.

---

## Referee Comment (RC2) · Anonymous Referee #2 · 1 Jun 2018

The manuscript presents the development of a high-performance computing capability (GCHP) for the GEOS-Chem CTM. The paper address relevant scientific modelling questions within the scope of GMD and presents a model with valid and clearly outlined methods.

It should in theory be possible for an independent scientist to construct a model that, while not necessarily numerically identical, will produce scientifically equivalent results. For the benchmarking, it should be possible for the results to be precisely reproduced.

[Figure]

**GMDD**

General Comments:

The large number of similar acronyms is confusing to the reader. Consider summarising in a table and perhaps showing the relation between the different modelling/data components by means of a diagram to aid the reader.

Beyond the scaling comparison between GCHP and GCC and changes in resoultion, a direct comparison between the two should be added (in the same resolution/configuration) to show the correctness but also the difference in accuracy and efficiency because of the new grid.

p.8 l.26: Any indication/reference on the error introduced by the correction?

p.12: Is the statement "The scalability of advection suffers from the additional communication overhead associated with reducing the domain size, as each domain must communicate a larger proportion of its concentration data to its neighbors" referring to the communication vs computation time for each MPI process?

Code Availability is outdated and has to be updated. Also, lisense information should be added.

Minor Comments:

p.8 l.13: Remove "information from"

p.8 l.18: "/" instead of "."

p.15 l.8: Remove "an unchanged copy of"

p.15 l.12 Please consider rephrasing to "GCHP was tested and shown to scale from six cores up to at least 540 [...]"

---

## Author Comment (AC1) · 28 Jun 2018

Dear Editor and Reviewers,

We thank the reviewers and the executive editor for their comments and suggestions on our manuscript. Please find attached a PDF which includes a full summary of the reviewer comments with responses from the authors, followed by two copies of the manuscript: first with all edits shown explicitly in red, and then a clean manuscript with all edits integrated.

[Figure]

Regards,

Sebastian Eastham

Please also note the supplement to this comment:
https://www.geosci-model-dev-discuss.net/gmd-2018-55/gmd-2018-55-AC1-supplement.pdf

———————————————————

---

## Author Response (AR1)

28th June 2018

Dear Editor,

Re: Review of "GEOS-Chem High Performance (GCHP): A next-generation implementation of the GEOS-Chem chemical transport model for massively parallel applications" for *Geoscientific Model Development*

We thank the referees for taking the time to review our paper, and for their insightful comments. We have listed their reviews in full below, with a response given for each point. Taking into account the recommendations from each reviewer and the comment from the executive editor, we have modified the manuscript (attached), highlighting any changes which were made since the original submission. We also attach a "clean" copy of the updated manuscript. Where possible, line numbers are included in the responses which correspond to the "clean" updated manuscript. In addition to the points raised by the reviewers and by the editor, we have made two other minor sets of changes. Firstly, the lower section of figure 5 (AOD) was previously and erroneously showing a one-hour rather than one-day average. This has been updated, and does not change the findings of the paper. Secondly, some minor changes have also been made to the text to improve clarity, but again these changes are superficial and all highlighted in the attached manuscript. We hope that you will find our edits satisfactory and look forward to your comments.

*Executive Editor's Comment*

Dear authors,

1. Please include the version number for GCHP in the title, and throughout the manuscript.

**Response: We have changed the title and several references to GCHP to instead read as "GCHP v11-02c".**

2. The precise version of the code discussed in the manuscript must be made available. The current best practice is for this code to be uploaded to a public repository and a DOI assigned. The DOI should be cited in the manuscript.

**Response: We have now uploaded the specific version of the code used here to a public repository (GitHub). This version of the code has been assigned a DOI, which is provided in the Code Availability statement.**

*Reviewer #1*

Summary

I am not sure what to expect from these sort of papers as they do describe complex systems with only couple of pages of words. So if you are an outside reader then it is almost impossible to get a firm idea about the described system. It is also debatable whether these brief descriptions have any usefulness to other researcher? The model described in this

paper has mainly two components, (i) a chemistry module, which is local and should, in principle, have a near perfect scaling (without any MPI instruction), while the second component (ii) is advection, which is not local and requires halo filling (or communication on demand) for parallel implementations. The only possible reason for publishing such papers would be to provide a reference for the model in question.

**Response: The paper is indeed by necessity too short to provide a complete description of the model. To address the reviewer's concern, we now provide links to the GCHP website where a detailed description of the model including a user's manual are available. As recognized by the reviewer, publication of this paper in GMD will provide an important reference for the very large community of GEOS-Chem users wishing to use GCHP. But it will also be useful to developers of other models who may want to follow a similar development strategy, and they can go to the GCHP website to learn more.**

Detailed comments

1. I think the paper needs a table summarizing the definitions of all the acronyms (GEOS-Chem, CTM, GCHP, ect...) used, because I kept going forward and backward to look for their meanings in the text.

**Response: A table has now been added to the introduction (Table 1) which lists every acronym used in the paper.**

2. My understanding of atmospheric chemistry models is that they solves a system of coupled ODEs at each grid of the model and each grid point is, in principle, independent of its neighbors. In other words, the chemical model needs information on the grid only and therefore such process should scale perfectly and these models are ideal for parallelism. However, advection needs information about the characteristic (hyperbolic problems) and this part that needs effort to make it work with parallel implementation (MPI). This looks like a coding task of combining FV3 transport (Lin et al.) with an existing chemistry model (with OpenMP). I think, the whole description of the new system and how its differs from the original GCC could be improved. There should be more clarifications and a detailed description of what changed from the original code (probably a table listing all the components of GCHP and how it differs from the original GCC).

**Response: One of the main features of GCHP is that it does not change the code from GCC, and in fact uses the exact same code to calculate local chemistry. We now stress this detail in section 2.2 ("GCHP v11-02c model architecture"): "Within each atmospheric domain, local terms are calculated by a standard copy of the GEOS-Chem Classic code, embedded in the model as described by Long et al. (2015). This copy of the GEOS-Chem code is identical to that used in GEOS-Chem Classic, such that all processes other than advection which are simulated in GCC are simulated identically in GCHP". We also now clarify that "The embedded copy of GEOS-Chem in GCHP is compiled without OpenMP shared-memory parallelization, resulting in a pure MPI implementation" (page 5, lines 3 to 9).**

3. Page 1, Abstract, Line 2. I am not sure how large the system of chemical species is? It would be better to give an order of unknowns for a typical system, i.e., order 100-1000000?.

**Response: We have added a clarification that these systems typically involve of the order of 100 - 1,000 chemical species (page 1, line 16).**

4. The discussion at page 7 in relation to semi-Lagrangian, cube-sphere and lat-long grids is a bit subjective!
   a) I am not sure what the authors means by "inconsistency in technique" in the sentence "The problem can be mitigated …...in ensuring mass conservation"?

**Response: GCC uses a mix of Eulerian and semi-Lagrangian methods, depending on the local CFL number. However, we agree that this is a minor point only and does not add significantly to the discussion. As such we have changed this list to focus on the question of mass conservation (page 7, line 9).**

   b) I think the whole passage "In an MPI environment, ….. sharing the pole" cannot be justified. Semi-Lagrangian (SL) schemes on any grid with the right halo size can achieve near perfect scaling [see for example with up to 100000 cores SL scalability experiments in the paper: *High-performance high-resolution semi-Lagrangian tracer transport on a sphere. J.B. White III and J.J. Dongarra, Journal of Computational Physics, Vol. 230, pp. 6778-679 (2011)*] or up to 10000 cores scalability results in *Allen and Zerroukat, Journal of Computational Physics, Vol. 319, p. 44-60 (2016)]*. Certainly the only argument against SL is the lack of inherent conservation, but in terms of accuracy, stability, simplicity, computational cost, and scalability there is no valid arguments against SL.

**Response: We agree that the problems here stem more from the use of a rectilinear lat-lon grid than from the use of the semi-Lagrangian method. We now clarify simply that the use of a semi-Lagrangian method when CFL > 1 results in increased domain halo size, and therefore increased communication (page 7, lines 10-12).**

   c) Of course there is no perfect scheme and every approach has some advantages and disadvantages and one has to be objective about these things. For example the cube-sphere has the disadvantages of dealing with 6 panels and their orientations and the non-orthogonality of the grid and its associated grid-imprinting (for example see your figure 5 for C24 where it is very clear that the pattern of the tracer distribution is very much influenced by the grid). It would be interesting to show the equivalent pictures (blowup pictures at Figure 5 for C24) due to the original GCC!.

**Response: We now include results from GCC simulations at two resolutions in our comparisons of simulated ozone and AOD. Accordingly, Figure 5 has been split into two. The new Figure 5 shows ozone at 4km when simulated at C24 and C180 (GCHP) as well as at 4°×5° and 2°×2.5° (GCC). Figure 6 now shows AOD for the same 4 simulations. Furthermore, we use 2-D histograms (Figure 7) to directly compare estimated ozone at 4 km between GCC (at 4°×5°) to the results at both C24 and C180 with GCHP (regridded to 4°×5°). We believe that this provides a convincing**

**demonstration of GCHP's ability to reproduce the capabilities of GCC while also highlighting some of the differences between the two architectures.**

5. Page 8, 1st paragraph of section 2.3.2. In parallel domain decomposition, I would suggest to use the right nomenclature of halo-filling instead of "providing boundary conditions", because there is no real boundaries between processors.

**Response: We have changed this line accordingly (page 8, line 10).**

6. Page 8, end of 2nd paragraph of section 2.3.2. It is not clear what does mean the comment: "GCHP defaults instead to a simple global air mass correction also applied to tracers"?? My understanding is that the main purpose of using FV3 is to achieve mass conservation in an inherent way so I don't understand what is this correction is doing? Does it mean the inherent conservation is lost? Further clarifications are needed here.

**Response: The greatest advantage of the FV3 model, as we perceive it, is to enable the execution of the model using a grid other than the rectilinear latitude-longitude grid previously in use. However, any CTM which uses time-averaged winds and interpolated surface pressure tendencies to estimate advection will inherently introduce a mass conservation error (Jöckel et al. 2001). Once mass flux information is available directly from the parent GCM or data assimilation system - in this case, the NASA GEOS DAS - this error is expected to be significantly reduced, but currently the standard meteorological products from all major forecasting centers do not include mass fluxes, instead reporting wind speeds. Until these fluxes are available, a mass correction is needed to ensure that tracer mass is conserved. In our case we have used a simple global mass correction, which has advantages and disadvantages as discussed in the aforementioned paper. We now clarify in the text that no pressure fixer has yet been designed for advection on the cubed sphere, but that GCHP is designed to accept mass fluxes once available (page 8, lines 22-25).**

7. I don't see any use for to the bottom sub-figure (tim/1000colms) of figure 3.

**Response: The bottom sub-figure is designed to show that the model's performance and scalability per computer core improve as the resolution is increased, due to the diminishing effect of the I/O overhead, and that for higher resolutions GCHP's performance per grid point is superior to that of GCC. This is now clarified explicitly in the paper on page 11, lines 8-11.**

8. Section 4 (example simulation). It is good to show how the scalability of GCHP can be exploited to run higher-resolution simulations and the improvement that comes as results of that (Figure 5). However, I feel there are some figures missing in this section/paper. Although there is a lot figures comparing GCC and GCHP in terms of computational efficiency (scaling and wall clock time), for fidelity there should be at least a couple of figures comparing (at least visually) GCC and GCHP for a benchmark test to show that the solutions of the two models are, at least visually, in a good agreement.

**Response: We agree completely that it is useful to compare the output of GCHP to that from GCC, in order to validate the results. As mentioned in a prior comment, we**

**have therefore expanded this section to include multiple comparisons between GCHP and GCC.**

*Reviewer #2*

The manuscript presents the development of a high-performance computing capability (GCHP) for the GEOS-Chem CTM. The paper address relevant scientific modelling questions within the scope of GMD and presents a model with valid and clearly outlined methods.

It should in theory be possible for an independent scientist to construct a model that, while not necessarily numerically identical, will produce scientifically equivalent results. For the benchmarking, it should be possible for the results to be precisely reproduced.

General Comments:

The large number of similar acronyms is confusing to the reader. Consider summarising in a table and perhaps showing the relation between the different modeling/data components by means of a diagram to aid the reader.

**Response: A table has now been added (Table 1) which clarifies all acronyms used in the table. Although Figure 1 shows the layout of GCHP diagramatically, we have also clarified on page 5 (lines 5-9) the relationship between pre-existing simulation code from GEOS-Chem (all columnar chemistry and physics), and new simulation code which is exclusive to GCHP (advection).**

Beyond the scaling comparison between GHCP and GCC and changes in resolution, a direct comparison between the two should be added (in the same resolution/configuration) to show the correctness but also the difference in accuracy and efficiency because of the new grid.

**Response: We have now expanded section 4 to include multiple comparisons between GCC and GCHP results, using both ozone and AOD at 4°×5° and 2°×2.5°.**

P.8 l.26: Any indication/reference on the error introduced by the correction?

**Response: The effect of these corrections was not quantified during the model runs, but the effect of using "pressure fixers" or mass corrections to counter the long-standing issue of inconsistency between winds and surface pressure tendencies is explored in detail in Jöckel et al (2001). In light of this comment and one from reviewer #1, we have extended the discussion of this correction to stress the advantage of using mass fluxes directly as opposed to inferring them from winds and surface pressure (page 8, lines 23-25).**

P. 12: Is the statement "The scalability of advection suffers from the additional communication overhead associated with reducing the domain size, as each domain must communicate a larger portion of its concentration data to its neighbors" referring to the communication vs computation time for each MPI process?

**Response: That is correct. Advection is the only simulated process which requires communication between processes.**

Code Availability is outdated and has to be updated. Also, lisense information should be added.

**Response: The Code Availability section has been updated in line with the recommendations of the Executive Editor, including the addition of a DOI. GEOS-Chem and GCHP both use the MIT License, and this detail has also been added to the Code Availability section (page 17).**

Minor Comments:

P.8 l.13: Remove "information from"

**Response: This sentence has been rewritten in response to comments from Reviewer #1 for clarity.**

P.8 l.18: "/" instead of "."

**Response: Corrected.**

P.15 l.8: Remove "an unchanged copy of"

**Response: Removed.**

P.15 l.12 Please consider rephrasing to "GCHP was tested and show to scale from six cores up to at least 540 [...]"

**Response: Rephrased accordingly.**

Thank you again for considering this work for publication in *Geoscientific Model Development*, and we would again like to thank the reviewers for their careful comments.

Sincerely,

Sebastian D. Eastham

[revised manuscript text omitted]